# TNF-α induces Claudin-1 expression in renal tubules in Alport mice

**Manami Iida[1], Shuichi Ohtomo[2], Naoko A. Wada[1], Otoya Ueda[1], Yoshinori Tsuboi[1], Atsuo Kurata[2], Kou-ichi Jishage[3], Naoshi Horiba[1]***

1 Research Division, Chugai Pharmaceutical Co., Ltd., Gotemba, Shizuoka, Japan, 2 Translational Research Division, Chugai Pharmaceutical Co., Ltd., Tokyo, Japan, 3 Chugai Research Institute for Medical Science Inc., Gotemba, Shizuoka, Japan

* horibanos@chugai-pharm.co.jp

**Data Availability Statement:** All relevant data are within the paper and its Supporting information files.

**Funding:** The author(s) received no specific funding for this work.

## Abstract

Claudin-1 (CL-1) is responsible for the paracellular barrier function of glomerular parietal epithelial cells (PEC) in kidneys, but the role of CL-1 in proximal tubules remains to be elucidated. In this study, to evaluate CL-1 as a potential therapeutic drug target for chronic kidney disease, we investigated change of CL-1 expression in the proximal tubules of diseased kidney and elucidated the factors that induced this change. We established Alport mice as a kidney disease model and investigated the expression of CL-1 in diseased kidney using quantitative PCR and immunohistochemistry (IHC). Compared to wild type mice, Alport mice showed significant increases in plasma creatinine, urea nitrogen and urinary albumin excretion. CL-1 mRNA was increased significantly in the kidney cortex and CL-1 was localized on the adjacent cell surfaces of PECs and proximal tubular epithelial cells. The infiltration of inflammatory cells around proximal tubules and a significant increase in TNF-α mRNA were observed in diseased kidneys. To reveal factors that induce CL-1, we analyzed the induction of CL-1 by albumin or tumor necrosis factor (TNF)-α in human proximal tubular cells (RPTEC/TERT1) using quantitative PCR and Western blotting. TNF-α increased CL-1 expression dose-dependently, though albumin did not affect CL-1 expression in RPTEC/TERT1. In addition, both CL-1 and TNF-α expression were significantly increased in UUO mice, which are commonly used as a model of tubulointerstitial inflammation without albuminuria. These results indicate that CL-1 expression is induced by inflammation, not by albuminuria in diseased proximal tubules. Moreover, we examined the localization of CL-1 in the kidney of IgA nephropathy patients by IHC and found CL-1 expression was also elevated in the proximal tubular cells. Taken together, CL-1 expression is increased in the proximal tubular epithelial cells of diseased kidney. Inflammatory cells around the tubular epithelium may produce TNF-α which in turn induces CL-1 expression.

## Introduction

Tight junctions (TJ) play an important role in sealing adjacent cells and forming epithelial barriers in various tissues, such as intestine and skin, to limit paracellular permeability. Claudin

**Competing interests:** The authors have declared that no competing interests exist.

(CL) is an important component of TJ and consists of 27 family members [1, 2]. CLs develop into various types of barriers depending on the expression patterns in distinct tissues. In skin, CL-1 is expressed in keratinocytes and form the skin barrier. The skin barrier prevents the invasion of foreign antigens and transepidermal loss of water. It has been suggested that a defect in CL-1 causes the disorder of the epidermal barrier and is related to the pathogenesis of atopic dermatitis and psoriasis [3, 4]. CL-1 has several functions in other tissues, e.g., CL-1 is expressed in liver and has been identified as a receptor for HCV infection [5, 6]. It has also been reported that increased CL-1 expression is responsible for the progression of cancer and inflammatory bowel disease [7, 8].

In kidneys, CL-1 has been reported to be expressed in glomerular parietal epithelial cells (PECs) and podocytes in mice [9]. A study using Zucker Diabetic rats indicated that CL-1 expression was increased in PECs along with proteinuria [10]. In a study using anti-glomerular basement membrane (GBM) mice, the reduction of CL-1 caused a decline in paracellular permeability in adjacent PECs. The authors suggested that CL-1 seals PECs by forming TJ and works as a second barrier to prevent the leakage of filtrated protein into the extraglomerular space [11]. Recently Samadi et al. reported that CL-1 expression was increased in streptozotocin-induced diabetic rats and that tropisetron, an antioxidative drug, improved renal function with suppression of CL-1 expression [12]. Moreover, it is reported that CL-1 expression is increased in crescent-forming PECs in human glomerulonephritis and research has been increasingly focused on the function of CL-1 in glomerulus [13]. On the other hand, the precise role of CL-1 expression in proximal tubules has not been revealed, though it has also been reported that the composition of CLs changes in diseased kidneys [14]. It has been shown that CL-1 expression is increased in the LPS-induced acute nephritis model [15], but also that albuminuria or hypotonic stress decreases CL-1 expression in tubular cells [16, 17]. Based on these contradictory studies, the role of CL-1 in proximal tubules in kidney disease remains controversial. In this study, to assess the potential of CL-1 as therapeutic drug target for CKD, we analyzed CL-1 expression in diseased proximal tubules in IgA nephropathy patients and in a chronic kidney disease (CKD) animal model and investigated the factors that induce this expression.

Alport syndrome, caused by mutations in Col4a3, Col4a4 or Col4a5, causes progressive kidney dysfunction with proteinuria by injuring the GBM [18]. The Col4a3 knock out mouse is a well-known as a model of Alport syndrome, characterized by proteinuria and progressive glomerular nephropathy with hyperplasia of GBM [19, 20]. Because Alport mice are also widely used to model of CKD [21], we also established an original strain of Alport mice. We confirmed that the mice exhibited kidney disorder and severe tubular dysfunction associated with glomerular injury. In this study, we investigated the change of CL-1 expression in Alport mice and explored the role and inducing factors of CL-1 in proximal tubule.

## Materials and methods

### Animals

Establishment of Col4a3 knockout mice: A Col4a3 gene knockout mouse strain was established by Zinc Finger Nuclease (ZFN)-mediated gene editing. A pair of ZFNs were designed to induce mutations in exon 48, which encodes the first part of the NC1 domain, a functional domain of the normal collagen chain (Fig 1A).

Preparation of UUO mice: Male C57BL/6J mice were obtained from Charles river (Kanagawa, Japan) at 8 weeks old. The UUO procedure was performed under isoflurane anesthesia as previously described [22]. Ligated kidneys were analyzed and non-ligated kidneys were used as control.

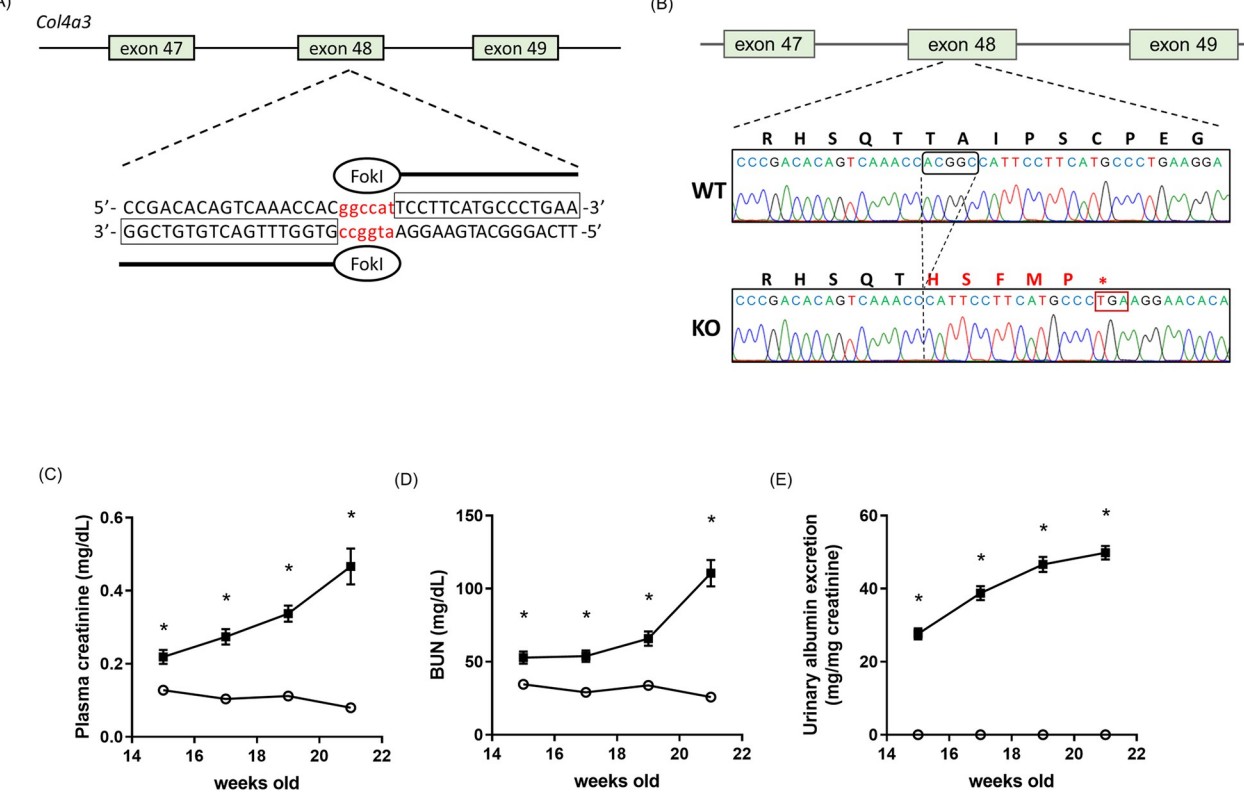

**Fig 1. Generation of a Col4a3 knockout mouse strain.** Schematic representation of the genomic structure of the exons 47 to 49 of mouse Col4a3 gene. ZFN binding sequences are printed in black and cutting site is printed in red. (A). Representative result of mutation analysis of the genomic PCR product by Sanger sequence analysis. WT, wild type. KO, Col4a3 knockout. Five nucleotides (ACGGC) deletion causes a premature terminal codon immediately by frame shift (B). The levels of plasma creatinine (C), BUN (D), and the urinary albumin excretion rate (E) were elevated in Alport mice. Black square indicates Alport mice and open circle indicates wild type mice as control. Data are represented as the mean ± SE (n = 5 in wild type mice, n = 8 in Alport mice). *p<0.05, significant difference versus wild type (Student's t-test).

All mice were maintained on a regular diurnal lighting cycle (12:12 light:dark) with free access to food (CE-2, CLEA Japan, Tokyo, Japan) and water. The maximum caging density was five mice in polycarbonate cage (173 x 278 x 134 mm). Woodchip was used as bedding and environmental enrichments including nesting material (paper clean, CLEA Japan) and shelter (Mouse Igloo, Bio-Serv, Flemington, NJ) were used. General conditions of animals were checked daily and body weight was measured twice a week to monitor health. During the studies, all animals kept healthy. Animal procedures and protocols were in accordance with the Guidelines for the Care and Use of Laboratory Animals at Chugai Pharmaceutical Co. Ltd., accredited by AAALAC and approved by the Institutional Animal Care and Use Committee (IACUC approval No. 18–440, 19–366).

## In vivo experimental design

Study 1: To analyze the progress of kidney disease in Alport mice, blood and urine samples were collected at 15, 17, 19, and 21 weeks old. Blood and urine samples were collected from the jugular vein, transferred into tubes with heparin, and centrifuged (13,000g, 15 min, 4°C) to prepare plasma samples. Urine samples were collected after blood sampling.

Study 2: Blood and urine samples were collected from Alport mice at 20 weeks old. Blood samples were collected from the abdominal portion of the vena cava under isoflurane anesthesia. Plasma samples were prepared just as in Study 1. Urine samples were collected on the day before sacrifice. After sacrifice by exsanguination under isoflurane anesthesia, kidney tissues were collected and dissected for pathology and mRNA expression analysis. Whole tissue was used for pathological analysis and the cortex and outer medulla were used for mRNA expression analysis.

Study 3: UUO mice were sacrificed by exsanguination under isoflurane anesthesia 7 days after surgery and kidney tissues were collected as described above.

### Biochemical analysis

Biochemical parameters in serum and urine were measured (creatinine, urea nitrogen, albumin (urine)) by using a TBA-120FR autoanalyzer (Canon Medical Systems Corporation, Tochigi, Japan).

### Pathological analysis

Kidney tissues were fixed with 10% neutral buffered formalin solution and then embedded in paraffin. The paraffin blocks were cut and stained with hematoxyline-eosin (HE) and periodic acid-Schiff (PAS) staining.

Immunohistochemistry was carried out on paraffin-embedded kidney tissue sections. Tissue sections were incubated in 10% normal goat serum in PBS for 30 minutes at room temperature to prevent non-specific binding. Kidney sections were incubated overnight at 4˚C with rabbit polyclonal anti-Claudin-1 (ab15098, Abcam, Cambridge, UK) followed by an anti-rabbit antibody for 30 min. Tissue sections were treated with 3,3'-diaminobenzidine (Dako kit, Agilent Technologies Inc., Santa Clara, CA). Sections were counterstained with Mayer hematoxylin.

Renal biopsy samples from patients with IgA nephropathy fixed in formaldehyde and embedded in paraffin (two males, 25 and 47 years old) were obtained from Avaden Biosciences (Seattle, WA). As a control kidney, a radical nephrectomy sample from a patient (50-year-old male) with renal cell carcinoma was also obtained from Avaden Biosciences. All procedures in this study were in accordance with the Declaration of Helsinki guidelines and approved by the Institutional Review Board of Chugai Pharmaceutical Co. Ltd.

### Gene expression analysis

RNA isolated from cells using a RealTime ready Cell Lysis Kit (Roche Diagnostics, Tokyo, Japan) or a quantity of 2 μg RNA isolated from kidney using a Tissue Lyser protocol (Qiagen, Hilden, Germany) were reverse transcribed with Transcriptor Universal cDNA Master (Roche Diagnostics). Quantitative PCR was performed with a TaqMan Gene Expression Assay (Thermo Fisher Scientific, Tokyo, Japan) on a LightCycler 40II (Roche Diagnostics). Expression levels are given as ratios to glyceraldehyde 3-phosphate dehydrogenase (GAPDH).

### In vitro experiments

RPTEC/TERT1 cells derived from human renal proximal tubules were purchased from American type culture collection® (ATCC, Manassas, VA). They were cultured in Dulbecco's modified eagle medium/F-12 medium (ATCC) and hTERT Immortalized RPTEC Growth Kit

(ATCC) in 5% CO2 at 37˚C. Cells were plated and rested overnight. Cells were incubated in Roswell park memorial institute 1640 medium (Nacalai Tesque, Inc., Kyoto, Japan) and 1% decomplemented fetal bovine serum for 24 hours before studies. Human serum albumin (Nacalai Tesque, Inc.) or recombinant human TNF-alpha protein (R&D Systems, Minneapolis, MN) were added at concentrations indicated in the figures. The mRNA and protein were purified from cells after 48 hours, followed by analysis by quantitative PCR and Western blotting, respectively.

## Western blotting

RPTEC/TERT1 cells or kidney tissue samples were homogenized in lysis buffer (Cell Signaling Technology, Danvers, MA) with protease inhibitor (Thermo Fisher Scientific). Protein concentrations were determined using bicinchoninic acid protein assay (Thermo Fisher Scientific). Western blotting was performed using Jess (Protein Simple, Inc., San Jose, CA). The primary antibody was rabbit polyclonal anti-Claudin 1 antibody (ab15098, Abcam) and rabbit polyclonal anti-beta Actin antibody (ab8227, Abcam). The secondary antibody was anti-rabbit immunoglobulin G (IgG) horseradish peroxidase (HRP)-conjugated antibody (31460, Thermo). Images were obtained using JesAmersham Imager 600 equipment (GE Healthcare, Chicago, IL) and were analyzed using Compass for SW (version: 4.0.0, ProteinSimple).

## Data and statistical analysis

For the in vivo studies, Student's t-test was performed. For the in vitro study, Dunnett's multiple comparisons test was performed. Statistical significance was set at p<0.05. Each statistical analysis was performed with GraphPad Prism (version 8.4.3; GraphPad Software, Inc., San Diego, CA)

## Results

### Establishment of Col4a3 knockout mouse strain

We designed a pair of zinc finger nuclease mRNAs that specifically target the nucleotide sequence in exon 48 of mouse endogenous Col4a3 (custom-designed by SIGMA-Aldrich Inc.) (Fig 1A). It is well known that the cDNA encompassing exons 48–52 codes the NC1 domain which is essential part of the triple helix structure of typical collagens. Several mutant mice were obtained by ZFN-mediated gene editing. Among these mutants, a mutant line having a 5 nucleotides (ACGGC) deletion in exon 48 was selected to establish a Col4a3 knockout mouse strain (officially named C57BL/6N-Col4a3em1Csk mice, referred to in this study as Alport mice) (Fig 1B and S1 Fig, S1 Method).

### Progression of kidney disease in Alport mice

Alport mice showed a significant increase in plasma creatinine, BUN, and the excretion of albumin into urine from 15 to 21 weeks old (Fig 1C–1E). Kidney dysfunction progressed gradually. We confirmed that our mice were similar to Alport mice that were previously reported [18, 19]. We decided to examine Alport mice at 20 weeks old because they showed severe kidney dysfunction at that age.

### CL-1 expression in Alport mice at 20 weeks old

The concentration of plasma creatinine, BUN, and the ratio of albumin to urine creatinine in Alport mice were significantly higher than in wild type mice and creatinine clearance was significantly declined in Alport mice compared with wild type mice at 20 weeks old (Fig 2A and

(A)

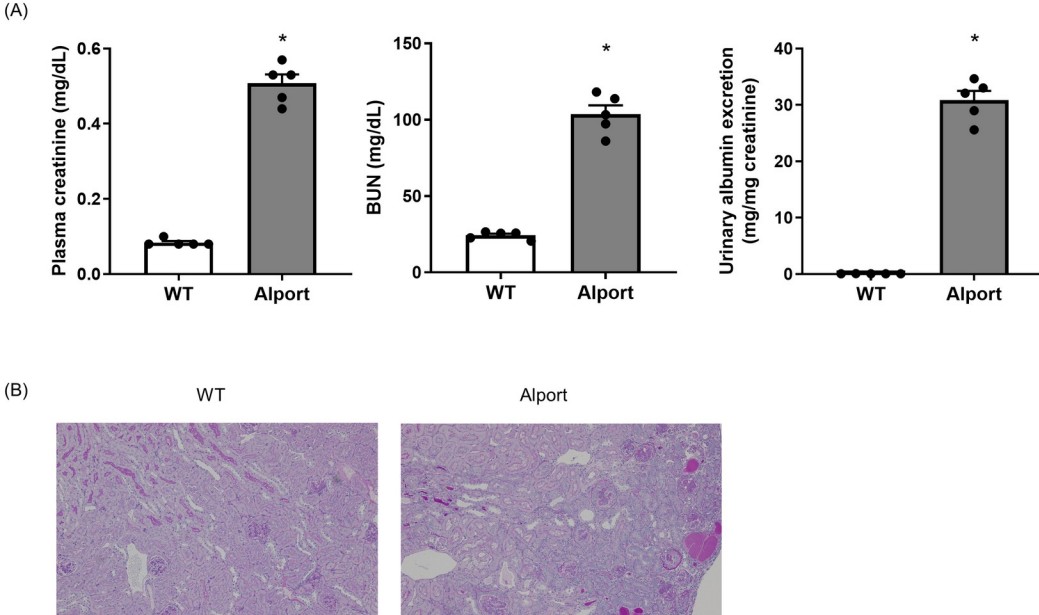

(B)

WT

Alport

**Fig 2. Kidney dysfunction in Alport mice at 20 weeks old.** Plasma creatinine level, BUN, and urinary albumin excretion rate were significantly higher in Alport mice than in wild type mice at 20 weeks old (A). Representative micrographs showing PAS stained kidney sections. Scale bars indicates 100 μm (B). The actual values are represented as dots and the columns represent mean + SE. *p<0.05, significant difference versus WT (Student's t-test). WT: wild type mice, Alport: Alport mice.

S2 Fig). Alport mice exhibited glomerulosclerosis, tubular dilation, the excretion of albumin into tubular lumens, and infiltration of inflammatory cells, especially mononuclear cells (Fig 2B).

CL-1 mRNA expression level in the kidney was significantly higher in Alport mice than in wild type mice (Fig 3A). In Alport mice, CL-1 protein was expressed both in tubular cells and PECs, while CL-1 protein was expressed mainly in PECs in wild type mice. In diseased kidney, a high level of CL-1 protein expression was seen especially between adjacent tubular cells, and severe mononuclear cells infiltration was confirmed around the tubule (Fig 3B). Inflammation-related molecules, TNF-α, and C-C motif chemokine ligand 2 (ccl2) mRNA were significantly increased and IFNγ mRNA tended to be higher in the kidney of Alport mice than in that of wild type mice (Fig 4).

### Increase of CL-1 expression by TNF-α in cultured tubular cells

We explored the factors inducing CL-1 using RPTEC/TERT1 cells. Because renal tubular cells are exposed to an excess of albumin in Alport mice, we examined the effect of albumin on CL-1 expression using RPTEC/TERT1 cells. Treatment with 100 μg/mL albumin changed neither CL-1 mRNA nor protein expression (Fig 5).

Next, because the marked infiltration of inflammatory cells was seen around high CL-1 expressing tubules and TNF-α mRNA level was increased in diseased kidney, we investigated the effect of TNF-α on CL-1 expression. Treatment with TNF-α increased the mRNA and protein levels of CL-1 dose-dependently in RPTEC/TERT1 cells (Fig 6).

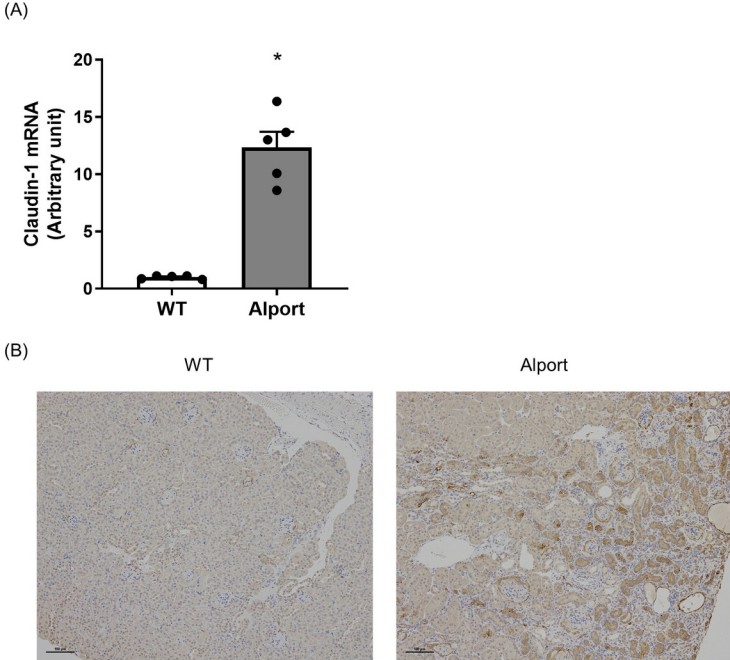

**Fig 3. Increased CL-1 expression in Alport mice.** In the kidney of Alport mice, CL-1 mRNA expression was significantly increased (A). Representative immunohistochemical images of CL-1 in the kidney. In Alport mice, CL-1 was strongly stained in proximal tubular cells. Scale bar indicates 100 μm (B). Relative mRNA expression was calculated as the ratio to GAPDH expression. The actual values are represented as dots and the columns represent mean + SE. $^*$p<0.05, significant difference versus WT (Student's t-test). WT: wild type mice, Alport: Alport mice.

## Increase of CL-1 expression in UUO mice

The UUO mouse model is widely used to model renal fibrosis without proteinuria. In UUO mouse kidney, mRNA expression of collagen type1α1 was significantly increased and remarkable fibrosis was identified by pathological analysis (S3 Fig). Renal CL-1 and TNF-α mRNA were significantly higher in UUO mice than in sham-operated mice. In addition, CL-1 protein expression was induced especially in proximal tubules in UUO mice as seen in Alport mice (Fig 7).

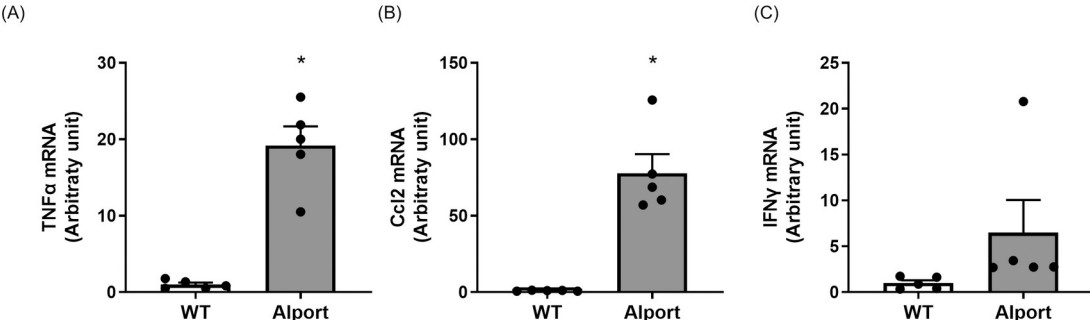

**Fig 4. Elevated expression of inflammation-related factors in the kidney of Alport mice.** Expression levels of TNF-α mRNA (A) and ccl2 mRNA (B) were significantly increased and IFNɤ mRNA expression tended to increase in the diseased kidney (C). Relative mRNA expression was calculated as the ratio to GAPDH expression. The actual values are represented as dots and the columns represent mean + SE. $^*$p<0.05, significant difference versus WT (Student's t-test). WT: wild type mice, Alport: Alport mice.

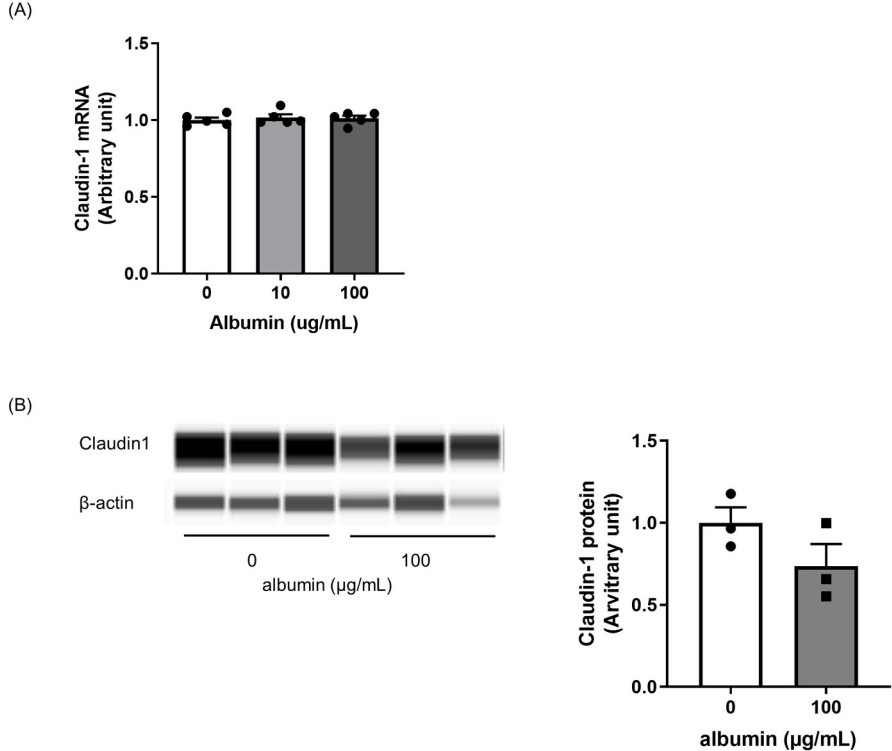

**Fig 5. No change in CL-1 expression in albumin-treated RPTEC/TERT cells.** RPTEC/TERT cells were treated with albumin for 48 hours. Albumin stimulation did not increase CL-1 mRNA expression (A) or protein (B). Relative mRNA expression and protein were calculated as the ratio to GAPDH and β-actin, respectively. The quantification of CL-1 was shown by the ratio to non-treatment cells. The actual values are represented as dots and the columns represent mean + SE.

## Expression of CL-1 protein in IgAN patients

We examined CL-1 expression in the kidneys of IgA nephropathy (IgAN) patients using IHC. In the kidneys of the healthy volunteer, CL-1 protein was predominantly expressed in PECs but not in renal tubules. While in the kidneys of IgAN patients, CL-1 expression was increased in PECs and also between adjacent renal tubular cells (Fig 8).

## Discussion

In the present study, we investigated the expression of CL-1 in diseased renal tubules using Alport mice as a CKD model. In wild type mice, CL-1 was expressed only in PECs, while CL-1 was expressed not only in PECs but also in adjacent proximal tubular cells in Alport mice. Therefore, we examined the mechanism of CL-1 upregulation using proximal tubular cells. Alport mice showed severe proteinuria and, in other kidney disease model mice, urinary albumin is reported to increase CL-1 expression in PECs [10], therefore we examined the effect of albumin on CL-1 expression in human proximal tubular cells. However, neither mRNA nor CL-1 protein showed any change and we concluded that albumin does not induce CL-1. We next examined the influence of inflammation on CL-1 expression because the infiltration of inflammatory cells was severe around proximal renal tubules showing high CL-1 expression. TNF-α is a major proinflammatory cytokine and is reported to increase TJ permeability by changing CL composition in various cell lines [23]. As in this study, TNF-α was significantly

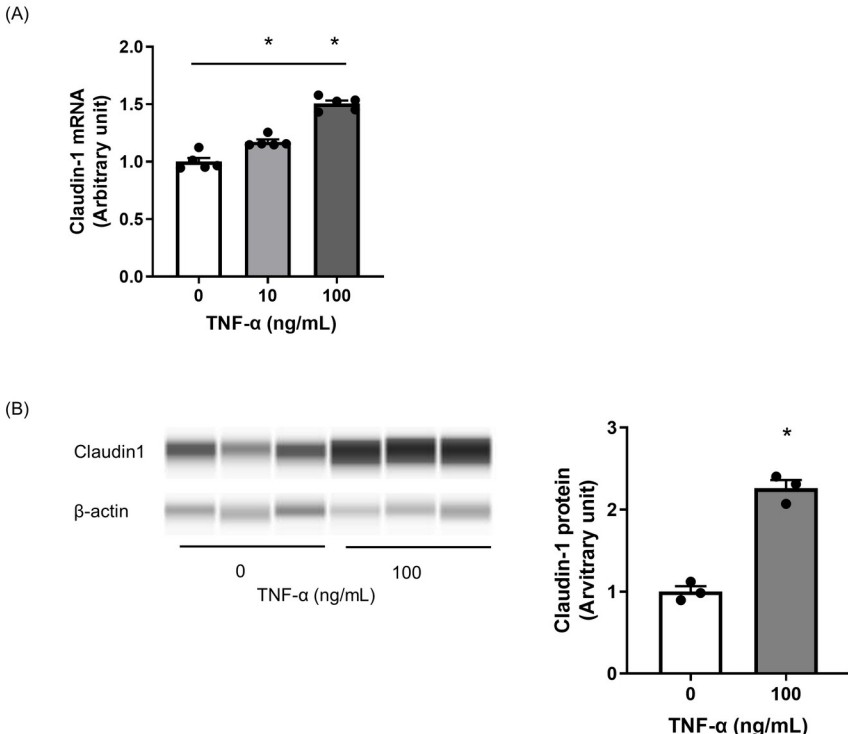

**Fig 6. Increase of CL-1 expression by TNF-α treatment in hTERT/RPTEC cells.** RPTEC/TERT cells were treated with TNF-α for 48 hours. CL-1 mRNA (A) and protein (B) expression levels were elevated by TNF-α treatment. Relative mRNA expression and protein were calculated as the ratio to GAPDH and β-actin, respectively. The quantification of CL-1 was shown by the ratio to non-treatment cells. The actual values are represented as dots and the columns represent mean + SE. *p<0.05, significant difference versus control (Student's t-test).

increased in Alport mice, so we examined the effect of TNF-α in RPTEC/TERT cells. Treatment with TNF-α increased CL-1 mRNA and protein expression dose-dependently. UUO mice, a model of renal inflammation and fibrosis without glomerular injury and proteinuria, also showed both elevated CL-1 expression in tubular epithelial cells and increased TNF-α gene expression. These results suggest that the increase in CL-1 expression is caused by inflammatory cells around the renal tubule, not by urinary albumin.

Our data showed that CL-1 was strongly expressed along the cell membranes of tubular cells in kidneys with IgA nephropathy in humans. Because the serum level of TNF-α is reported to be correlated with the severity of IgA nephropathy [24], it is possible that CL-1 expression is also induced by TNF-α in humans. Considering that TNF-α is also reported to be associated with lupus nephritis and diabetic nephropathy [25, 26], it is possible that the induction of CL-1 in renal tubules is a common feature in kidney diseases characterized by inflammation and fibrosis.

The reason for CL-1 upregulation in diseased renal tubular cells remains unknown. One possibility is the strengthening of the paracellular barrier. Amoozadeh et al. have reported that TNF-α increases both CL-1 expression and transepithelial resistance in the pig proximal tubular cell line LLC-PK1 [27]. Therefore, TNF-α is thought to strengthen the TJ barrier of renal tubular cells through CL-1 expression. In kidney disease, glomerular destruction causes the excretion of a large amounts of molecules to urine, which causes inflammation when it leaks from the renal tubules into interstitial space. At the same time, inflammatory cells induce CL-1

(A)

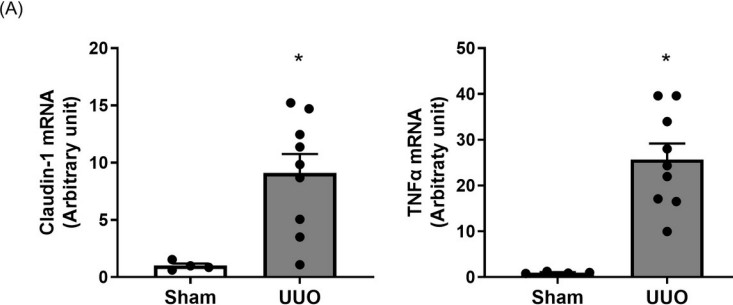

(B)

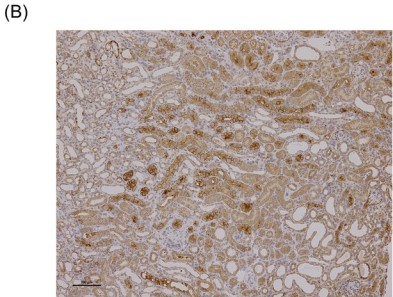

**Fig 7. Elevation of CL-1 and TNF-α in the kidney of UUO mice.** The mRNA expression levels of CL-1 and TNF-α were significantly elevated in UUO mice (A). Representative immunohistochemical images of CL-1 in the kidney of UUO mice. CL-1 was strongly stained in proximal tubular cells. Scale bar indicates 100 μm (B). Relative mRNA expression was calculated as the ratio to GAPDH expression. The actual values are represented as dots, and the columns represent mean + SE. *p<0.05, significant difference versus sham (Student's t-test). Sham: sham operated mice, UUO: UUO mice.

expression to strengthen the tubular epithelial barrier to prevent further leakage into the tubulo-interstitium.

There is another hypothesis regarding the role of CL-1 in injured tubules. In several studies, CL-1 has been reported to promote proliferation and epithelial-mesenchymal transformation (EMT). Bhat et al. have reported that TNF-α stimulation upregulated CL-1 expression in colon cancer cells and promoted cellular proliferation and migration [28]. It is also reported that CL-1 contributes to cellular proliferation and transformation in other cancer cell lines. The overexpression of CL-1 activated the cAbl-Ras-Raf1-ERK1/2 signal pathway which facilitated the expression of Slug and Zeb1 and led to the induction of EMT [29, 30]. In our current study, CL-1 was up-regulated especially in the proximal tubules, which contain more cells than normal tubules. This suggests that the elevation of CL-1 by TNF-α in tubular epithelial cells promotes cellular proliferation through the activation of cAbl, as also seen in cancer cells. Recently, injury-induced intratubular EMT has been reported to induce tubular atrophy and kidney fibrosis [31]. Therefore, the inhibition of EMT through the suppression of CL-1 expression may be a potential therapy target for kidney disease.

Though we revealed that CL-1 expression changes in diseased kidney, we could not determine whether CL-1 acts as a protective factor or a disease inducing factor in diseased tubules in the present study. To clarify this, we need to evaluate the effect of CL-1 suppression in kidney disease using siRNA or tubule-specific CL-1 conditional KO mice.

In conclusion, we identified that CL-1 expression was increased in renal proximal tubules both in Alport mice and IgAN patients and that CL-1 is induced by TNF-α due to

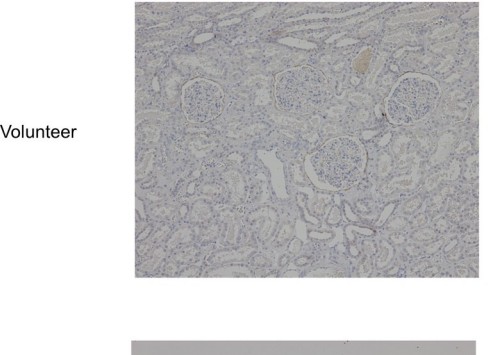

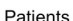

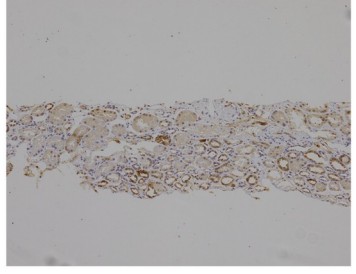
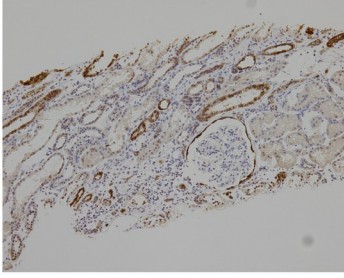

**Fig 8. Increase of CL-1 expression in IgAN kidney.** CL-1 expression was assessed using human kidney samples by IHC. CL-1 was expressed in PEC in kidney from a volunteer with renal cell carcinoma. In kidney from IgAN patients, CL-1 was expressed not only in PEC but also in proximal tubules.

inflammation. Further investigation of the role of CL-1 in kidney disease is needed to determine whether CL-1 can be a therapeutic target.

## Supporting information

**S1 Method. Establishment of Col4a3 knockout mice.**
(DOCX)

**S1 Fig. Generation of a Col4a3 knockout mouse strain.** (A) Schematic representation of the genomic structure of the exons 47 to 49 of mouse Col4a3 gene. Arrows show the location of primers used for PCR reaction followed by sequence analysis. (B) Representative agarose gel electrophoresis of the PCR product. Specific amplification around the exon 48 of mouse Col4a3 was confirmed by a single band at approximately 0.6 kb.
(TIF)

**S2 Fig. Creatinine clearance in Alport mice.** Creatinine clearance was declined in Alport mice at 20 weeks old. The actual values are represented as dots and the columns represent mean + SE. *$p < 0.05$, significant difference versus sham (Student's t-test). WT: wild type mice, Alport: Alport mice.
(TIF)

**S3 Fig. Infiltration of inflammatory cells and fibrosis in the kidney of UUO mice.** The mRNA expression level of collagen 1a1 was significantly elevated in UUO mice (A). Representative micrographs showing HE stained kidney sections. Scale bars indicates 100 μm (B). Relative mRNA expression was calculated as the ratio to GAPDH expression level. The actual values are represented as dots and the columns represent mean + SE. *$p < 0.05$, significant difference versus sham (Student's t-test). Sham: sham operated mice, UUO: UUO mice.
(TIF)

**S1 Raw image. Kidney dysfunction in Alport mice at 20 weeks old.** Individual micrographs showing PAS stained kidney sections. WT: wild type mice, Alport: Alport mice.
(TIF)

**S2 Raw image. Increased CL-1 expression in Alport mice.** Immunohistochemical images of CL-1 in the kidney. In Alport mice, CL-1 was strongly stained in proximal tubular cells. WT: wild type mice, Alport: Alport mice.
(TIF)

**S3 Raw image. Elevation of CL-1 in the kidney of UUO mice.** Immunohistochemical images of CL-1 in the kidney of UUO mice. CL-1 was strongly stained in proximal tubular cells. Sham: sham operated mice, UUO: UUO mice.
(TIF)

**S4 Raw image. Infiltration of inflammatory cells and fibrosis in the kidney of UUO mice.** Micrographs showing HE stained kidney sections. Sham: sham operated mice, UUO: UUO mice.
(TIF)

**S5 Raw image.**
(PDF)

**S1 Data.**
(XLSX)

## Acknowledgments

We thank Kaoru Matsumoto, Chisato Goto, Masahiro Morita, Toshiyuki Oshima, Hiromi Tateishi, Kanako Hara, Mami Kakefuda, Kiyoharu Sato and Yosuke Kawase for establishing and breeding the Alport mice, and Jacob Davis of Chugai Pharmaceutical Co., Ltd. for his assistance with English usage.

## Author Contributions

**Conceptualization:** Manami Iida, Shuichi Ohtomo, Kou-ichi Jishage, Naoshi Horiba.

**Data curation:** Manami Iida, Shuichi Ohtomo, Naoko A. Wada, Otoya Ueda, Yoshinori Tsuboi, Atsuo Kurata.

**Formal analysis:** Manami Iida, Naoko A. Wada, Otoya Ueda, Yoshinori Tsuboi, Atsuo Kurata, Kou-ichi Jishage, Naoshi Horiba.

**Investigation:** Manami Iida.

**Methodology:** Manami Iida.

**Writing – original draft:** Manami Iida, Naoko A. Wada.

**Writing – review & editing:** Otoya Ueda, Naoshi Horiba.

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
