## [Decision Letter · Decision Letter 0]

14 Dec 2021

PONE-D-21-25424Inflammation-mediated Claudin-1 protein induction in renal tubulesPLOS ONE

Dear Dr. Horiba,

Thank you for submitting your manuscript to PLOS ONE. After careful consideration, we feel that it has merit but does not fully meet PLOS ONE’s publication criteria as it currently stands. Therefore, we invite you to submit a revised version of the manuscript that addresses the points raised during the review process.

We look forward to receiving your revised manuscript.

Kind regards,

Franziska Theilig

Academic Editor

PLOS ONE

Journal Requirements:

Reviewers' comments:

Reviewer's Responses to Questions

**Comments to the Author**

1. Is the manuscript technically sound, and do the data support the conclusions?

Reviewer #1: No

Reviewer #2: Partly

2. Has the statistical analysis been performed appropriately and rigorously? 

Reviewer #1: Yes

Reviewer #2: Yes

3. Have the authors made all data underlying the findings in their manuscript fully available?

Reviewer #1: Yes

Reviewer #2: Yes

4. Is the manuscript presented in an intelligible fashion and written in standard English?

Reviewer #1: Yes

Reviewer #2: No

5. Review Comments to the Author

Reviewer #1: 1- The clinical value of this study should be pointed out in the introduction.

2- The authors should be used this reference in introduction or discussion which is related to claudin-1 in the kidney. PMID: 33047279

3- Which part of kidney used for gene expression or histological analysis and …?

4- The rationale behind this research work should be stated clearly in the “Abstract and Introduction” sections of the manuscript.

5- The IHC pictures should be clearly provided, they are blured.

6- Why the authors did not measure clearance in animals?

7- The authors shoud mention the anesthesia condition of animals ex, drug? Dose?....

8- Why did the authors select 7 days in their experiment?

Reviewer #2: Main Findings: The paper submitted by Iida et al., " Inflammation-mediated Claudin-1 protein induction in renal tubules ” investigates the change of CL-1 expression and factors that induce this change in proximal tubules of the diseased kidney. The authors used Alport mice as a kidney disease model and investigated the expression of CL-1 in diseased kidneys. The authors found that CL-1 mRNA was increased significantly in the kidney cortex, and CL-1 was localized on the adjacent cell surfaces of PECs and proximal tubular epithelial cells. Infiltration of inflammatory cells in the areas around proximal tubules and a significant increase in TNF-α mRNA were observed in diseased kidneys. The authors also used human proximal tubular cells (hTERT/RPTEC) to study the effect of TNF-α on CL-1 expression. The authors conclude that CL-1 expression is increased in the proximal tubular epithelial cells of the diseased kidney, and inflammatory cells around the tubular epithelium may have a role to play. The manuscript is interesting, with a lot of data to understand the mechanisms involved. The methodology is detailed, and the results are well explained.

Minor Comments:

1. The title needs some modification as it does not fit well with the study.

2. The western blot images need a lighter exposure to make the changes visible.

3. In figure 4C, the authors have shown increased IFN-γ mRNA in Alport mouse model but have not anywhere in the manuscript its role in changing CL-1 expression as this could also increase CL-1 expression being a pro-inflammatory cytokine.

4. I want authors to show the changes in other claudins, not only claudin 2 ( Figure 4B), as many other claudins are involved in inflammatory conditions.

5. Under the results section (Expression of CL-1 protein in IgAN patients), lines 275, 276 and 277, the authors have written the following sentences “The Results section should describe the most important findings of the study, analysis, or experiment. The most important results should be indicated, and relevant trends and patterns should be described” which I could not understand and seems a copy-paste from somewhere.

6. The authors suggested that the increase in CL-1 expression is caused by inflammatory cells around the renal tubule but have not shown what kind of inflammatory cells anywhere in the study. These results need to be shown as the whole paper is based on inflammation.

7. The discussion needs rewriting as some of the statements contradict their findings.

8. The paper needs major editing for English and grammatical errors.

6. PLOS authors have the option to publish the peer review history of their article (what does this mean?). If published, this will include your full peer review and any attached files.

Reviewer #1: **Yes: **yes

Reviewer #2: **Yes: **Ajaz Ahmad Bhat

---

## [Author Response · Author response to Decision Letter 0]

2 Feb 2022

Dear Dr. Theilig and reviewers,

We are pleased to submit a revision of our manuscript retitled as “TNF-α induces Claudin-1 expression in renal tubules in Alport mice” (PONE-D-21-25424). We thank you for your careful reading of our manuscript and comments. We revised our manuscript according to your suggestions and have expressed our opinion in the response letters. 

We believe the revised manuscript will satisfy your requests.

We look forward to hearing from you and thank you again for your careful consideration our manuscript.

Kind regards,

Naoshi Horiba

---

## [Decision Letter · Decision Letter 1]

23 Feb 2022

TNF-α induces Claudin-1 expression in renal tubules in Alport mice

PONE-D-21-25424R1

Dear Dr. Horiba,

We’re pleased to inform you that your manuscript has been judged scientifically suitable for publication and will be formally accepted for publication once it meets all outstanding technical requirements.

Kind regards,

Franziska Theilig

Academic Editor

PLOS ONE

Additional Editor Comments (optional):

Reviewers' comments:

Reviewer's Responses to Questions

**Comments to the Author**

1. If the authors have adequately addressed your comments raised in a previous round of review and you feel that this manuscript is now acceptable for publication, you may indicate that here to bypass the “Comments to the Author” section, enter your conflict of interest statement in the “Confidential to Editor” section, and submit your "Accept" recommendation.

Reviewer #1: All comments have been addressed

Reviewer #2: All comments have been addressed

2. Is the manuscript technically sound, and do the data support the conclusions?

Reviewer #1: Yes

Reviewer #2: Yes

3. Has the statistical analysis been performed appropriately and rigorously? 

Reviewer #1: Yes

Reviewer #2: Yes

4. Have the authors made all data underlying the findings in their manuscript fully available?

Reviewer #1: Yes

Reviewer #2: Yes

5. Is the manuscript presented in an intelligible fashion and written in standard English?

Reviewer #1: Yes

Reviewer #2: Yes

6. Review Comments to the Author

Reviewer #1: the manuscript improved more. all comments addressed. congratulations to the authors. It is now acceptable.

Reviewer #2: I am satisfied with the revision. The authors have done a good job in revising the manuscript and I do not have any further comments so my recommendation is "Accept"

7. PLOS authors have the option to publish the peer review history of their article (what does this mean?). If published, this will include your full peer review and any attached files.

Reviewer #1: No

Reviewer #2: **Yes: **Ajaz A. Bhat

---

## [Editor Report · Acceptance letter]

2 Mar 2022

PONE-D-21-25424R1 

TNF-α induces Claudin-1 expression in renal tubules in Alport mice 

Dear Dr. Horiba:

I'm pleased to inform you that your manuscript has been deemed suitable for publication in PLOS ONE. Congratulations! Your manuscript is now with our production department. 

Kind regards, 

on behalf of

Dr. Franziska Theilig 

Academic Editor

PLOS ONE